# Genetic Variability, Character Association, and Path Coefficient Analysis in Transplant Aman Rice Genotypes

**DOI:** 10.3390/plants11212952

**Published:** 2022-11-02

**Authors:** Abu Salah Muhammad Faysal, Liakat Ali, Md. Golam Azam, Umakanta Sarker, Sezai Ercisli, Kirill S. Golokhvast, Romina Alina Marc

**Affiliations:** 1Department of Genetics and Plant Breeding, Faculty of Agriculture, Bangabandhu Sheikh Mujibur Rahman Agricultural University, Gazipur 1706, Bangladesh; 2Plant Breeding Division, Bangladesh Agricultural Research Institute, Gazipur 1701, Bangladesh; 3Department of Horticulture, Faculty of Agriculture, Ataturk University, 25240 Erzurum, Turkey; 4Siberian Federal Scientific Center of Agrobiotechnology RAS, 2b Centralnaya, 630501 Krasnoobsk, Russia; 5Food Engineering Department, Faculty of Food Science and Technology, University of Agricultural Sciences and Veterinary Medicine, 400372 Cluj-Napoca, Romania

**Keywords:** rice (*Oryza sativa*), genetic diversity, heritability, genetic advance, correlation, path coefficient

## Abstract

A field experiment was carried out with 20 genotypes of Transplant Aman (T. Aman) rice at the Department of Genetics and Plant Breeding, Bangabandhu Sheikh Mujibur Rahman Agricultural University, Salna, Gazipur-1706, Bangladesh. The study was performed to evaluate the genetic deviation, trait association, and path coefficient (PC) based on grain yield (GY) and different yield-contributing agronomic characters. Variance analysis displayed extensive traits-wise variations across accessions, indicating variability and the opportunity for genetic selection for desirable traits. The high mean, range, and genotypic variances observed for most of the characters indicated a wide range of variation for these traits. All the characters indicated the minimum influence of environment on the expression of the trait and genetic factors had a significant role in the expressivity of these characters. High heritability in broad sense (h^2^_b_) and high to moderate genetic advance in percent of the mean (GAPM) were recorded for all the characters except for panicle length (PL). Based on mean, range, and all genetic parameters, the selection of all the traits except for PL would contribute to the development of T. Aman rice genotypes. A correlation study revealed that selection based on plant height (PH), number of effective tillers per hill (NET), PL, number of filled spikelets per panicle (NFS), flag leaf length (FLL), spikelet sterility (SS) percentage, and harvest index (HI) would be effective for increasing the GY of rice. Genotypic correction with grain yield (GCGY), PC and principal component analysis (PCA) revealed that direct selection of NFS, HI, SS%, and FLL would be effective for improving the GY of rice in future breeding programs.

## 1. Introduction

Rice (*Oryza sativa* L.) is the most dynamic main food for one-third of the world’s people, and rice production in Asia accounts for nearly 90% of the total output [1]. The production of rice, the oldest and the greatest significant crop, contributes not only to the provision of food security but also to the generation of money and job opportunities [2]. Worldwide, the rice cultured area surpasses 16.2 crore ha and total produce of 75.5 crore tons [3]. Bangladesh has long been known as the world’s fourth biggest rice producer, after only India, Indonesia, and China. The rice culture occupies 75% of all agricultural land in the country and employs 48% of the nation’s farm laborers. It contributes 70% of the agricultural gross domestic product (GDP), while its proportion of national revenue is 1/6 [4]. Roughly 11 million hectares are dedicated to rice farming, a number that has remained almost unchanged for more than the last three decades and which produces approximately 3.4 crore tons [4]. Rice is still the dominant source of energy, fiber, minerals, and protein in the Asian diet. Of the entire daily calorie intake, >1/2 of the dietary calories and a lion’s share of protein come from it. The use of variation in genotypic level for improving rice production potential is a critical way to feed over 9 billion people by 2050 and adapt to climate change.

In Bangladesh, rice is planted year-round, with three separate growing seasons: Boro, Aman, and Aus. It is cultivated in four ecosystems, viz., irrigated rice (Boro), rainfed or partially irrigated (transplanted Aus and Aman), rainfed upland (direct-seeded Aus), and deep-water (broadcast Aman) [5]. Considering total production, Aman is the country’s second-largest crop after Boro rice. Broadcast Aman is planted in lowlands from mid-March until mid-April, then transplanted in July and August. Approximately 46 lacs ha of rain-fed Aman rice plantations in Bangladesh provide >50% of the country’s total rice output [6]. In certain agro-ecological situations, T. Aman becomes the major crop, promoting the large-scale growth of T. Aman rice.

Genetic richness in any germplasm is vital for any crop improvement effort since it is the key to integrating favorable alleles and bringing about desired modifications [7,8]. Breeding and agricultural development require a thorough understanding of current genetic diversity [9,10]. Inborn deviation among traits is central to bringing and selecting anticipated types [11,12]. For increasing grain output via breeding, it is important to recognize the species’ variability, the nature of character associations, and the involvement of various traits [13,14]. The growing demand for rice necessitates the development of outstanding genotypes that can thrive in an assortment of environments [15,16]. Grain yield (GY) is a multifaceted attribute that is impacted by an arrangement of genes, the environment in which they develop, and the degree and types of diversity in genotypes [17]. In addition, GY is directly or indirectly interrelated with other agronomic traits such as plant stature, growth spell, panicle length (PL), tiller per plant, loaded grains per panicle, and primary and secondary branches per panicle [18,19]. The primary focus of plant breeders concentrates on picking desirable features in the blend, granting each an economic gain based on GY to generate a preference index [20]. 

Despite GY having a low genetic inheritance, it is possible to improve yields via many yield-related component attributes [21]. Selecting traits only based on heredity may result in a bad option, while combining genetic advancement and heritability may be more advantageous [22]. Knowing the association of yield and related causal variables is essential for most efforts to discover plant selection guidelines [23,24]. Furthermore, GY and associated characteristics may also be partitioned into direct and indirect impacts to find which characteristics are most responsible for increasing seed yield [25]. Therefore, it is crucial to link component qualities to yield and to each other. Research on rice GY and various component properties has been investigated phenotypically in a vast number of studies [18,26]. There is a positive significant association of GY with days to maturity, grains per panicle, days to 50% flowering (DFF), number of effective tillers per hill (NET), PL, number of filled spikelets per panicle (NFS), and spikelet fertility (SS) [17,18,27].

Path coefficient (PC) analysis subdivides the genetic link of GY with its paying traits into direct and indirect PCs, revealing useful features as standards for selection to boost the GY of rice [1,7]. It evaluates the reason for the link of two variables based on simple correlations among characters and linearity and additively. It may be used to determine each character’s contribution to the total yield. The current research studied genetic diversity, heritability, genetic advance, association, and contribution of attributes to GY, and PC analysis is specifically chosen genotypes to provide selection criteria for high-yielding rice varieties [8,25].

Therefore, the present study was undertaken to study the variability among T. Aman rice genotypes, determine the relative contributions of each character towards yield and their inter-relationships, and find out the direct and indirect PCs of component traits towards GY.

## 2. Results

### 2.1. Genetic Variability

There were substantial deviations in the accessions for all traits (Table 1). The extent of variation in GY and the related agronomic variables that contribute to GY were examined and the mean value, range, genotypic variance (σ^2^_g_), phenotypic variance (σ^2^_p_), environmental variance (σ^2^_e_), genotypic coefficient of variation (GCV), phenotypic coefficient of variation (PCV), heritability in broad sense (h^2^_b_), genetic advance (GA), and genetic advance in percent of the mean (GAPM) are shown in Table 2. An extensive genetic deviation was recorded for DFF among the genotypes, which suggested a considerable variation for this character (Table 1). DFF varied from 95 to 113 days (Table 2). Phenotypic variance (σ^2^_p_) for DFF (26.39) was slightly superior to the σ^2^_g_ (25.98). As revealed by the characteristic, the trait’s expressivity was significantly impacted by genetic rather than environmental variables. PCV (4.87) was close to the GCV (4.83), suggesting a minimal environmental effect on the manifestation of the trait. The estimate of h^2^_b_ for this variable was high (98.46), while the GA (10.42) and GAPM (9.87) were moderate, indicating that apparent variation was owing to genotypes.

ANOVA for plant height (PH) showed a highly noteworthy genotypic mean sum of squares (Table 1). These differences exposed a large spectrum of variability for PH across genotypes. PH ranged from 111.20 to 177.00 cm, with a mean of 137.75 cm (Table 2). The σ^2^_p_ (371.63) was only marginally superior to σ^2^_g_ (368.90) for this trait, indicating a minimal environmental effect on the expressivity of the genes regulating the characteristic. The deviation of PCV (13.99) and GCV (13.94) was minimum (Table 2) for PH. PH exhibited a high h^2^_b_ (99.26) connected to high GA (39.42) and GAPM (28.62), indicating that apparent variation was attributed to genotype.

NET ranged from 6.50 to 10.20. The σ^2^_p_ (0.84) was a little bit higher than σ^2^_g_ (0.78) and PCV (10.96) was close to the GCV (10.61) (Table 2). Thus, the trait had low contextual influence, but genetics played a significant influence on NET expressivity. The GA was low (1.77), but h^2^_b_ of NET was high (93.77) along with GAPM (21.16). Noteworthy variations in flag leaf length (FLL) were found among the tested genotypes (Table 1). The genotypes showed a large matrix of variation from 25.04 to 40.41 cm (Table 2). σ^2^_p_ (19.28) was the closest to σ^2^_g_ (19.24). The deviation of PCV (13.68) and GCV (13.67) showed a less environmental influence on FLL (Table 2). The inheritance of FLL was high (99.83), with a high GAPM (28.14). However, GA was low (9.03). PL ranged from 22.35 to 28.90 cm, with a mean of 26.43 cm (Table 2). There was a visible deviation in σ^2^_p_ (1.48) and σ^2^_g_ (0.84) and PCV (4.61) and the GCV (3.48), indicating a considerable environmental influence on the manifestation of the PL (Table 2). PL showed high h^2^_b_ (82.93) but very low GA (1.43) and GAPM (5.42).

The number of primary branches per panicle (PBP) showed low σ^2^_p_ (1.09) and σ^2^_g_ (0.82) and moderate h^2^b (75.25) and low GA (1.62), indicating a considerable environmental influence on PBP (Table 2). However, the number of secondary branches per panicle (SBP) had high h^2^_b_ (89.14) was high with low GA (9.35) with higher GAPM (26.52). The NFS ranged from 76 to 162, with a mean of 129.58. σ^2^_p_ (502.57) was superior to σ^2^_g_ (482.64) and had close disparities between PCV (17.30) and GCV (16.95), with high h^2^_b_ (96.03) revealing NFS has a sturdy genetic regulator. Like NFS, the genetic parameters of SS percentage also indicate a strong genetic control among the selected lines.

The 1000-grain weight (TGW) ranged from 21.95 to 31.42 g, with a mean of 27.10 g. σ^2^_p_ (8.37), σ^2^_g_ (8.33), and the deviation of PCV (10.67) and GCV (10.65) indicate the least environmental influence on the character expression (Table 2). h^2^_b_ of TGW was very high (99.49), with a significant GAPM (21.88). There is a large deviation in harvest index (HI) among the genotypes (28.24 to 37.88). σ^2^_p_ of HI (8.56) was very close to σ^2^_g_ (8.49). In addition to h^2^_b_ (99.19), PCV (9.02), GCV (8.98), and GAPM showed that HI had more genetic control than the environment. The genotypes differ significantly for GY. GY ranged from 4.55 tons to 8.18 tons, with a mean of 6.61 tons (Table 2). GCV (13.60), PCV (14.39), and h^2^_b_ (89.40) showed that the environment has less influence on the GY of selected rice genotypes.

### 2.2. Character Association

DFF had insignificant positive correlation (IPC) with NET (0.018, 0.016), SS (0.048, 0.048), and TGW (0.187, 0.187), respectively at both genotypic and phenotypic (G and P) level (Table 3 and Table 4). DFF and TGW proved a positive connection, indicating that late-flowering cultivars yielded panicles with higher weights. Inversely, DFF had an insignificant negative correlation (NC) with PH, FLL, PL, PBP, SBP, NFS, HI, and GY per hectare at both levels. PH showed significant positive correlation (SPC) with FLL (r = 0.719, 0.718), PL (r = 0.730, 0.644), HI (r = 0.498, 0.497), and GY (r = 0.587, 0.580) at G and P levels, respectively, suggesting that yield might be augmented by raising PH. Plant stature showed an IPC with NET, SBP, NFS, and TGW. Alternatively, this variable exhibited an NC with PBP (r = −0.066, −0.062) and SS (r = −0.293, −0.092) at the G and P levels, respectively.

NET was found to parade a SPC with NFS (r = 0.621, 0.617), HI (r = 0.505, 0.501), and GY (r = 0.656, 0.635) at both the G and P levels, respectively. Therefore, yield could be enhanced by augmenting NET. SS showed a negative and noteworthy interrelation (r = −0.447, −0.440) with NET at both levels, which indicated an increase in NET resulted in a decrease in the SS percentage. The correlation between NET and the rest of the characters was insignificant. The FLL a showed an SPC with PL (r = 0.723, 0.647) and GY (r = 0.472, 0.463) at the G and P levels, respectively, indicating that PL and GY could be increased by improving this trait. However, the NC between FLL and SS (%) was insignificant but considerable. The results indicated that an increase in FLL decreased the SS percentage (Table 3 and Table 4). The association of this trait with PBP, SBP, NFS, and HI was insignificant in the positive direction.

PL had a SPC with NFS (r = 0.451), at a phenotypic level, but with HI (r = 0.529, 0.476) and GY (r = 0.576, 0.492), both at the G and P levels. This result specifies that GY was increased with the increase of PL and HI. SS showed a negative and significant genotypic relationship (r = −0.455) with PL, which indicated that an increase in PL decreased the SS percentage. The association of this trait with PBP, SBP, and TGW was insignificant in a positive direction.

PBP showed an insignificant NC with TGW, HI, and GY both at the G and P levels. This result indicates that PBP increases and decreases the GY, along with HI, and TGW. PBP showed an IPC with SBP, NFS, and SS percentages. The negative association between GY and PBP indicated that the assortment based on this appeal would not be concrete for bumping GY. In contrast, the SBP showed a significant positive connection with NFS (r = 0.548, 0.538) at both G and P levels, indicating that NFS could be increased by improving SBP. SBP showed an insignificant relationship in a positive direction with HI and GY. Even though this positive relationship was small but still important, the results showed that HI and GY could be raised by raising SBP.

NFS displayed a highly momentous positive link with GY (r = 0.488, 0.485) and HI (r = 0.611, 0.595), both at the G and P levels, respectively, indicating that GY increased with the increase of NFS and HI. SS (r = −0.625, −0.620) and TGW (r = −0.627, −0.623) correlated negatively with NFS at both levels, indicating that they adversely reacted to this character. The SS exhibited a very significant NC with GY (r = −0.646, −0.634) and negative and insignificant interrelation with HI (r = −0.349, −0.348) at both G and P levels, respectively, indicating that increasing SS lowered HI and GY. The trend of association between TGW and GY and HI was positive but trivial both at G and P levels. Therefore, TGW had little influence on GY for the genotype studied. The association between HI (r = 0.866, 0.845) and GY was extremely positive both at the G and P levels, showing HI might boost grain production.

### 2.3. PC Analysis

Path analysis was utilized to discover the link between GY and other yield-donating traits, allowing for a deeper understanding of the interaction. In the current inquest, GY per hectare was measured as a resultant (dependent) variable and DFF, PH, NET, FLL, PL, PBP, SBP, NFS, SS (%), TGW, and HI were causal (independent) variables. Using the genotypic PC analysis, the indirect and direct PCs (bold phase) of eleven causative factors on GY have been illustrated in Table 5.

DFF displayed direct PC in a positive direction (0.018) on GY (Table 5). While the indirect PC was found to be negative via FLL (−0.015), NFS (−0.103), SS (−0.011), and HI (−0.101). Indirect PCs were positive for PL (0.007), PBP (0.013), SBP (0.019), and TGW (0.049), which were almost negligible in magnitude. As a result, the overall correlation (−0.125) between GY and other features turned negative. Consequently, increasing rice GY could not be achieved by selecting DFF. Despite the direct PC in a negative direction of PH (−0.011) on GY, FLL (0.110), TGW (0.106), and HI (0.237) had strong indirect PCs in a positive direction, consequentially a statistically SPC with GY (0.587) (Table 5). However, the positive PCs through DFF (0.004), NET (0.006), PBP (0.015), and SS (0.061) were minimal. The indirect PC is balanced out by several other factors, which makes the overall association of GY and PH (0.587) positive.

The NET had a negligible direct PC in a positive direction (0.015) on GY. The indirect PC in a positive direction via NFS (0.334) and HI (0.238) on GY were sizable, triggering a high SPC (0.656). The positive impacts of PH (0.003), FLL (0.040), PBP (0.035), and SS (0.091) were trifling. The indirect PCs were negative for PL (−0.004), SBP (−0.046), and TGW (−0.056), which were almost negligible in magnitude. Therefore, improving grain production may be accomplished either by directly selecting NET or indirectly selecting through NFS. The FLL bore a direct PC and positive (0.150) on GY. It did not have much of a PC on PH (−0.008), PL (−0.019), PBP (−0.007), or SBP (−0.030). The indirect impact via DFF (0.001), NET (0.005), NFS (0.062), SS (0.046), TGW (0.070), and HI (0.202) on GY was positive, which caused the overall positive correlation (0.472) between FLL and GY.

PL had a trivial adverse direct PC (−0.021) on GY. The indirect PCs of this trait on yield via NFS (0.196) and HI (0.208) were positive and considerable, making the total SPC (0.576). Whereas the positive PCs via DFF (0.014), PH (0.011), NET (0.021), FLL (0.098), and SS (0.086) were marginal. The PBP (−0.014) and SBP (−0.041) had a negative and practically insignificant stimulus on GY. As a result, there was an SPC between PL and GY (0.576).

PBP and SBP had a direct negative (−0.209 and −0.209) impact on GY, respectively. The indirect PC of PBP on yield via FLL (0.004) and NFS (0.137) was positive but almost negligible in magnitude. PBP showed indirect PCs in negative direction through DFF (−0.004), PH (−0.002), NET (−0.004), PL (−0.009), SBP (−0.036), SS (−0.005), TGW (−0.056), and HI (−0.117). As a consequence, the total correlation (−0.303) was negative with GY. The indirect PCs of SBP on yield via DFF (0.003), PH (0.004), NET (0.007), FLL (0.029), NFS (0.284), and HI (0.204) were positive. Indirect PCs in negative direction were lean via PL (−0.007), PBP (−0.028), SS (−0.007), and TGW (−0.0722). The interrelation between SBP and GY was positive but small (0.216), hence direct selection for SBP will not boost GY.

NFS showed the highest positive direct PC (0.542) on yield (Table 5, and Figure 1). A considerable indirect PC in a positive direction via HI (0.232), causing a total correlation significantly positive (0.611), whereas the PCs in positive direction via DFF (0.001), PH (0.002), NET (0.010), FLL (0.021) and SS (0.125) were low. The indirect PCs of NFS on GY via PL (−0.009), PBP (−0.049), SBP (−0.103), and TGW (−0.161) were negative and almost negligible in magnitude. As a result, improving GY by simply relying on this NFS is a possibility. The SS showed a direct PC in a negative direction (−0.200) on GY. The indirect PCs of PH, PL, and TGW in a positive direction were smaller than the indirect PCs in a negative direction via DFF, NET, FLL, PBP, SBP, NFS, and HI, resulting in the overall correlation being negative and very significant (−0.646). Therefore, direct selection based on SS could be effective for increasing GY. The TGW had a positive direct impact on GY (0.268) and a positive indirect impact on GY through DFF (0.004), FLL (0.037), PBP (0.042), SBP (0.061), and HI (0.015). Indirect negative impacts through PH (−0.005), NET (−0.002), NFS (−0.333), and SS (−0.082) reduced the overall association (0.004). The result revealed that direct selection of TGW would have little influence on GY. HI showed a direct PC in a positive direction (0.479) on GY (Figure 1). The indirect PC of HI on GY via NFS (0.263) was positive, resulting in the total correlation being highly SPC (0.866). Thus, selection directly based on this character would be achievable for increasing GY.

### 2.4. Principal Component Analysis (PCA)

The biplots also visualized the position of the 20 accessions and illustrated the correlations among traits of this study (Figure 2). Principal component analysis (PCA) of yield and yield-contributing traits of 20 genotypes generated 12 PC, and the first two components together explained more than 50 percent of the total variation (Figure 2 and Appendix A). PC1 contributed 37.5% of the total variation, which was mainly represented by the variation in PH, NET, FLL, PL, and HI. However, SS is the only major contributing trait that was negatively associated with GY. PC2 accounted for 18.8% of the variation and represents mostly the variation in PH, FLL, and TGW. The PCA revealed that the NFS, GY, FLL, PH, and HI were closely associated and were the main contributors in PC1.

## 3. Discussion

The GY is a complicated product impacted by many interdependent quantitative factors. Therefore, it is probable that unless other factors affecting yield are taken into account, selection for yield may not be effective [1]. The improvement of one character that is closely linked to yield has a knock-on effect on several other closely connected characteristics [1]. In other words, the plant breeder has a blueprint for selection if they understand how other traits impact yield and how they interact. Using this information, plant breeders can better understand how genetic and non-genetic components interact [28].

Success in crop breeding schemes is largely determined by genetic diversity and the transmission of desired characteristics. Exploring the genetic diversity of a species with the intervention of plant breeders may result in the improvement of commercially desired qualities [1]. An indispensable trait is yield, which emerges from the multiplicative interactions of several contributing factors [29]. To provide food security for the anticipated expanding population, breeders have boosted the production capacity of all key crops. To address these complex relationships, we must create high-yield varieties by introgression traits from existing genotypes with high trait values. In the current study, the genotypes disclosed highly significant variation for all the traits (Table 1). To investigate the genetic diversity of any genetic resources, G and P variations, coefficient of variation as well as its h^2^_b_, GA, were frequently exploited [29,30]. The mean and range for all the traits studied specified the presence of gigantic variability for the GY and its allied traits, which provides more opportunity to utilize these traits for the further rice improvement program. The amount of heterogeneity in germplasm might be influenced by σ^2^_g_ and σ^2^_p_ of traits [30]. 

All studied genotypes displayed significant deviations across traits which specified a large array of variability among the traits. Similar deviations were also displayed in morphological traits [31,32,33,34,35], and other yield-contributing traits of different crops [20,36,37]. The selection of genotypes based on σ^2^_p_ may be deceptive owing to environmental impacts. Hence, splitting the σ^2^_p_ into genotypic and environmental effects is crucial [38]. NFS, PH, SS, DFF, SBP, and FLL had higher σ^2^_g_, suggesting that these characteristics had more variability and, thus, a wider window of opportunity for selection [1]. The σ^2^_p_ was somewhat greater than the σ^2^_g_, showing that the phenotypic manifestation of these features was influenced by the environment to some extent. There have been several studies in the past that documented environmental impacts on yield-contributing characteristics of rice [27,38,39]. The differences between σ^2^_p_ and σ^2^_g_ were very small for DFF, NET, FLL, SS, and TGW, indicating additive gene action for these traits. This observation is consistent with the previous research findings [40] that reported minimal differences between σ^2^_p_ and σ^2^_g_ for PH, NFS, SS, and HI.

High PCV and GCV values suggest that there is a great amount of genetic diversity, whereas traits with low PCV and GCV values illustrate a low level of diversity [1,8]. The presence of considerable variation implies the prospect of successful selection for character enrichment. Environmental influences may be seen in the lower GCV and higher PCV values. In the present research, SS had the greatest GCV and PCV, followed by NFS, PH, FLL, SBP, and GY. This suggests that GY might be improved by using these features as a selection criterion. The result is in line with the previous research that reported that the GY of rice could be improved through the selection of these traits [1,7]. A considerably lower GCV and PCV were found for PL, PBP, and DFF, indicating that selection would not be effective based on these traits. This result was comparable with the previously reported results [1]. Heritability in the broad sense was classified as high (>60%), moderate (40–60%), and low (40%) [41]. Among the 12 traits studied, 11 exhibited high heritability, except PL which displayed moderate heritability indicating higher possibility of selection for the improvement of rice. The levels of GA were ranked as low, moderate, and high, with corresponding ranges of 10%, 10–20%, and >20%, respectively [42]. High h^2^_b_ coupled with high GA of a trait is crucial for the selection of a trait in any breeding program. In the present study, high h^2^b coupled with high GAPM was reported for PL, NET, FLL, SBP, NFS, SS, TGW, and GY, suggesting the additive gene action for the traits, so selection based on these traits could be contributed largely for the improvement of rice. Previous literature reported high heritability with high genetic advance in percent of the mean for different yield-related traits [1]. PBP had high h^2^b with moderate GAPM (13.30), indicating the equal influence of additive and non-additive gene action in the manifestation of the traits. Similar results were reported by several researchers on rice [1]. The calculated variability, h^2^_b_, and GAPM among the genotypes tested indicate that there is plenty of room for selection to improve rice GY.

The selection of desirable phenotypes based only on yield may not be realistic due to the sensitivity of quantitative traits to environmental impact [43]. In addition to h^2^_b_ and GA, to boost yield genetically, selection should be based on the correlation coefficient analysis of GY and its component characters [1,14]. For achieving the highest yield potential, all quantitative factors that influence yield must be considered [1]. The association between GY and yield-related characteristics facilitates the identification of traits that, via indirect selection, may boost seed yield [1]. The prime economic trait, GY, showed significant positive G and P correlation at (*p* ≤ 0.01) with HI, NFS, PH, PL, NET, and FLL. A similar finding was reported by other researchers [1,7,27]. Pratap et al. [39] reported that GY had a strong positive association with SS, filled grains panicle^−1^, NET, PH, and TGW in rice. Fentie et al. [3] found an SPC between GY with NFS, HI, PL, FLL, and PH. Li et al. [2] reported a positive, significant correlation that persisted between GY and field grain per panicle, TGW, PH, and PL. In the present study, DFF, PBP, and SS were negatively correlated with GY. However, a significant negative phenotypic and genotypic correlation was found between GY and SS. The NC of these variables with GY makes concurrent selection profitable implying that, GY would be high with a low SS percentage. The result was consistent with the previous studies, which reported an NC between DFF, NPB, and SS of rice [1,18,44].

PC analysis offers a realistic assessment of the direct and indirect stimulus of each trait linked with the other traits [43]. PC analysis has been used to assess selection criteria in a vast array of species. Using path analysis, the G and P correlation coefficients were further segmented into direct and indirect impacts through substitute features or pathways. The principles of such analysis create inklings to the encounters of important component traits useful in the indirect selection of multifaceted traits such as yield [1]. In the present study, based on stepwise selection methods of regression, 11 traits were selected as causal variables for GY. GY was considered as a resultant variable and DFF, PH, NET, PL, PBP, SBP, SS, TGW, and HI were causal (independent) variables. PC analysis of the genotypic correlations revealed that NFS had the highest positive direct PC on GY, followed by HI, and TGW. GY also had a direct PC in a positive direction on DFF, NET, and FLL. The positive relationship between NET, FLL, and GY emphasizes the need to directly select these attributes to boost rice seed production. The results were similar to the results of Takele et al. [45] who revealed that HI and DFF had a direct PC in a positive direction on GY. Hasan-Ud-Daula and Sarkar [1] reported that NFS had a direct PC in a positive direction on the GY of rice [1]. Roy et al. [46] reported that there was a significant direct PC in a positive direction of TGW and HI on the GY of T. Aman rice. PBP, SBP, SS, PH, and PL had a direct negative impact on GY. These results are consistent with results reported by Hasan-Ud-Daula and Sarkar [1], who found a direct PC in a negative direction of SS, SBP, and PH on GY. In addition to the direct and indirect PCs of HI, NFS also had a positive correlation with GY. These results are in accordance with the findings of other research that reported HI, and NFS had a considerable indirect PC in a positive direction on GY [47]. Therefore, selection focused on these traits would be gratifying for the GY enhancement of rice. The residual (0.13) indicates that the traits included in the genotypic path analysis can explain 87% of the total variation of GY. The least residual effect (0.13) of the genotypic path for GY indicated that all the important characters were included in the present study. The principal component analysis reflects that NFS, TGW, FLL, and HI were the main contributing traits for yield. According to the findings of this study, traits with a direct PC in a positive direction and a positive and extensive assembly with GY should be given special consideration in the selection process. The residual effect was 0.13 indicating that 87% of the variability was explained by the 12 characters studied. There were other contributors (13%) which were responsible for yield but were not taken into consideration in the present investigation. 

## 4. Materials and Methods

### 4.1. Experimental Location and Source of Germplasm

The research was carried out on the Bangabandhu Sheikh Mujibur Rahman Agricultural University’s (BSMRAU) experimental farm between the years 2015 and 2016. Investigations were conducted out in the Madhupur Tract (AEZ28, about 24°23′ N–90°08′ E, with a mean elevation of 8.4 m above the mean sea level [1]. A typical rice-growing lowland with a soil pH of 6.5 was used in the experiment. The test area’s particular rainfall pattern makes it an ideal place for T. Aman rice cultivation, with a rainy summer and a mainly dry winter. The soil exhibited low levels of organic matter (0.87 percent), total N (0.09 percent), and exchangeable K (0.13 cmol/kg) [1]. A total of twenty T. Aman rice genotypes from the central seed gene bank of BSMRAU were employed in the experiments. These advanced lines were developed by hybridizing between local and modern inbred lines or cultivars of T. Aman rice. The rice genotypes are presented in Appendix A.

### 4.2. Experimental Layout and Observation Recording

The experiment was conducted using a Randomized Complete Block Design (RCBD) with three replications. A single genotype was planted in a 5.0 m × 1.0 m plot. A seedling at 30 days of age [48] was shifted per hill, with a 20 cm × 20 cm space between rows and plants. Each genotype consists of 5 rows with 25 hills per row, for a total of 125 hills per replication. To keep the soil fertility in the experimental units at the optimal level for rice cultivation in Bangladesh, additional chemical fertilizers including urea, tripolyphosphate, muriate of potash (KCl), and gypsum (CaSO_4_·2H_2_O) were sprayed at rates of 220-120-90-60 kg/ha, respectively [1]. Three applications of total urea were administered at 15, 30, and 45 days after transplantation. During the growing season, all the important intercultural operation (gap filling, weeding at regular intervals, insecticide application, top dressing of fertilizers, etc.) was done to help the plants grow and develop healthily. Statistical information was gathered from ten indiscriminately chosen hills for each genotype under each replication from the middle two rows to avoid border effects. Data were collected based on DFF, PH (cm), NET, TGW (g), FLL (cm), PL (cm), (PBP), (%) (SS), (SBP), NSP, NFS, HI, and GY (t/ha).

### 4.3. Statistical Analysis

Every character was subjected to an analysis of variance (ANOVA) using the PROC GLM model in SAS version 9.2 for Windows. Through the use of SPSS, estimates were made for the mean, range, and standard deviation of each characteristic. The mean sum of squares (MS) of error and σ^2^_p_ were estimated following Johnson et al. [42]. The square root of the error mean was used to get the error variance (σ^2^_e_). The G and P components of variance (σ^2^_g_ and σ^2^_p_), respectively, phenotypic and genotypic coefficients of variation (PCV, GCV) [49], broad-sense heritability (h^2^_b_) [50], genetic advance (GA) [42], and genetic advance as percent of the mean (GA%) [51] were estimated by adapting the formulae suggested by the researchers were mentioned in the table (Table 6). Correlation coefficients were further partitioned into components of direct and indirect effects by PC analysis [52]. PCA was conducted by factoextra package in R version 4.3.1.

## 5. Conclusions

The σ^2^_p_ and PCV were slightly higher than the corresponding σ^2^_g_ and GCV for all the characters indicating the minimum influence of the environment on the expression of the trait and genetic factors had a significant role in the expressivity of these characters. Except for PL, all the characters showed high to moderate mean and range, moderate PCV, GCV, high h^2^_b_, and high to moderate GAPM indicating selection of these characters could significantly contribute to improving the yield potential of rice. These results revealed that superior genotypes could be used as preferable cultivars and moderate yielder genotypes could be used as parent materials in future hybridization programs for the development of new cultivars because of the presence of high heritability and GAPM. The genotypic coefficient of correlation was generally a little bit higher than the corresponding phenotypic coefficient of correlation which indicated that the apparent association might be due to genetic reasons. SS (%) showed a direct PC in a negative direction. As an NC was desirable for SS (%), direct selection through these characters would be effective for improving the GY of rice. Although GCGY for PH, NET, PL, NFS, FLL, SS (%), and HI were significant but considering three analyses, such as GCGY, PC and PCA revealed that direct selection of NFS, SS (%), FLL, and HI would be effective for improving the GY of rice. The residual effect was 0.13 indicating that 87% of the variability was explained by the 12 characters studied. Therefore, the present study suggested that NFS, SS (%), FLL, and HI which are the main components of the yield of these genotypes should be given high priority in selection for future breeding programs.

## Figures and Tables

**Figure 1 plants-11-02952-f001:**
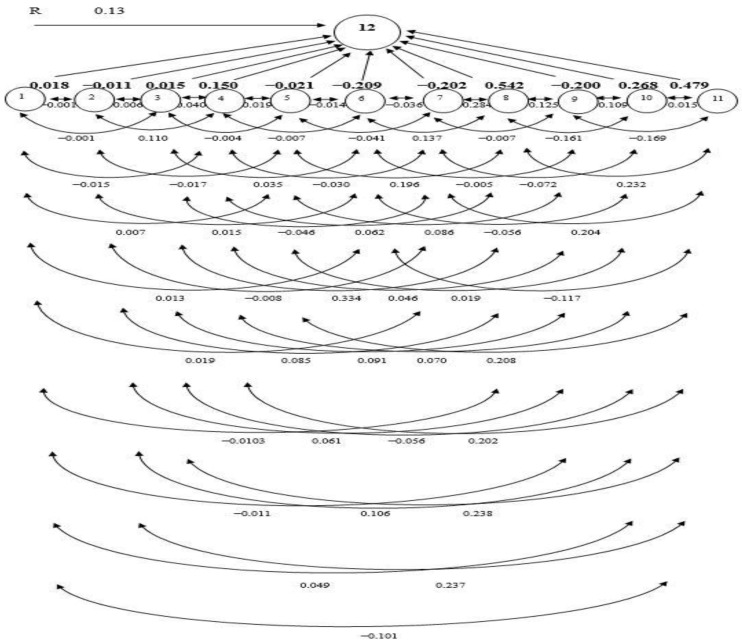
The path coefficient (PC) diagram representing cause and effect relationships among quantitative traits and grain yield. (1. days to 50% flowering; 2. plant height; 3. number of effective tillers per hill; 4. flag leaf length; 5. panicle length; 6. number of primary branches per panicle; 7. number of secondary branches per panicle; 8. number of filled spikelets per panicle, 9. spikelet sterility, 10. 1000-grain weight; 11. harvest index; 12. Grain yield (t/ha); R—residual effect).

**Figure 2 plants-11-02952-f002:**
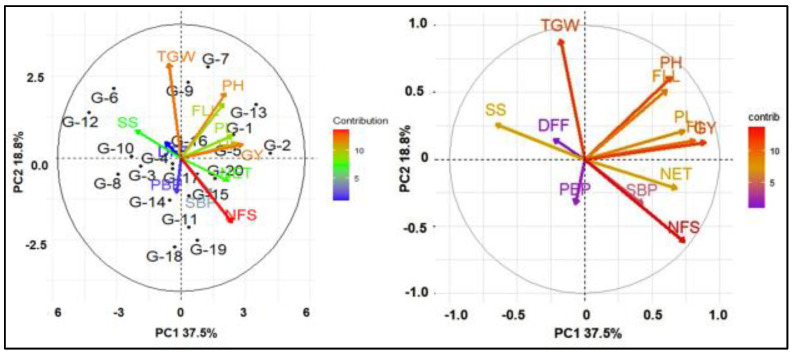
Biplot based on PCA shows the relationship of secondary traits with grain yield. DFF—days to 50% flowering; PH—plant height; NET—number of effective tillers per hill; FLL—flag leaf length; PL—panicle length; PBP—number of primary branches per panicle; SBP—number of secondary branches per panicle; NFS—number of filled spikelets per panicle; SS—spikelet sterility; TGW—1000-grain weight; HI—harvest index; GY—grain yield.

**Table 1 plants-11-02952-t001:** Analysis of variance (mean sum of squares) for yield and its component characters in 20 T. Aman rice genotypes.

Sources of Variation	df	DFF	PH (cm)	NET	FLL (cm)	PL (cm)	PBP	SBP	NFS	SS (%)	TGW (g)	HI (%)	GY (t/ha)
Genotypes	19	78.36 **	1109.43 **	2.41 **	57.77 **	3.18 **	2.73 **	72.23 **	1467.85 **	286.17 **	25.02 **	25.55 **	2.52 **
Replication	2	0.26	2.46	0.03	0.01	0.60	0.15	7.12	22.82	0.81	0.18 *	0.04	0.17
Error	38	0.41	2.74	0.05	0.03	0.64	0.27	2.82	19.94	0.26	0.04	0.07	0.10
CV (%)		0.60	1.20	2.73	0.57	3.02	4.62	4.76	3.45	2.26	0.76	0.81	4.68

df—degree of freedom; DFF—days to 50% flowering; PH—plant height; NET—number of effective tillers per hill; FLL—flag leaf length; PL—panicle length; PBP—number of primary branches per panicle; SBP—number of secondary branches per panicle; NFS—number of filled spikelets per panicle, SS—spikelet sterility, TGW—1000-grain weight; HI—harvest index; GY—grain yield; CV—Coefficient of variation, * Indicates significant at 5% level and ** Indicates significant at 1% level.

**Table 2 plants-11-02952-t002:** Estimation of genetic parameters for yield and its component characters in 20 genotypes of T. Aman rice.

Sources of Variation	DFF	PH (cm)	NET	FLL (cm)	PL (cm)	PBP	SBP	NFS	SS (%)	TGW (g)	HI (%)	GY (t/ha)
Mean	105.53	137.75	8.36	32.09	26.43	11.30	35.27	129.58	22.72	27.10	32.46	6.61
Range	95.00–113.00	111.20–177.00	6.50–10.20	25.04–40.41	22.35–28.90	9.00–13.00	25.00–45.00	76.00–162.00	7.97–52.15	21.95–31.42	28.24–37.88	4.55–8.18
σ^2^_g_	25.98	368.89	0.78	19.24	0.84	0.82	23.14	482.64	95.30	8.33	8.49	0.81
σ^2^_p_	26.39	371.63	0.84	19.28	1.48	1.09	25.95	502.57	95.56	8.37	8.56	0.90
σ^2^_e_	0.40	2.73	0.05	0.03	0.63	0.27	2.82	19.94	0.26	0.04	0.07	0.09
GCV	4.83	13.94	10.61	13.67	3.48	8.02	13.64	16.95	42.97	10.65	8.98	13.60
PCV	4.87	13.99	10.96	13.68	4.61	9.24	14.44	17.30	43.03	10.68	9.02	14.39
h^2^_b_	98.46	99.26	93.78	99.83	57.10	75.25	89.14	96.03	99.72	99.49	99.19	89.40
GA	10.42	39.42	1.77	9.03	1.43	1.62	9.36	44.35	20.08	5.93	5.98	1.75
GAPM	9.87	28.62	21.16	28.14	5.42	14.33	26.52	34.22	88.39	21.88	18.42	26.49

σ^2^_g_—genotypic variance; σ^2^_p_—phenotypic variance; σ^2^_e_—environmental variance; GCV—genotypic co-efficient of variation; PCV—phenotypic coefficient of variation; h^2^_b_—heritability in broad sense; GA—genetic advance; GAPM—genetic advance in percent of mean; DFF—days to 50% flowering; PH—plant height; NET—number of effective tillers per hill; FLL—flag leaf length; PL—panicle length; PBP—number of primary branches per panicle; SBP—number of secondary branches per panicle; NFS—number of filled spikelets per panicle, SS—spikelet sterility, TGW—1000-grain weight; HI—harvest index; GY—grain yield.

**Table 3 plants-11-02952-t003:** The genotypic correlation coefficient among 12 characters in 20 genotypes of T. Aman rice.

	PH	NET	FLL	PL	PBP	SBP	NFS	SS	TGW	HI	GY
DFF	−0.036	0.018	−0.090	−0.284	−0.067	−0.112	−0.193	0.048	0.187	−0.212	−0.125
PH		0.202	0.719 **	0.730 **	−0.066	0.060	0.151	−0.293	0.384	0.498 *	0.587 **
NET			0.235	0.345	−0.197	0.258	0.621 **	−0.447 *	−0.237	0.505 *	0.656 **
FLL				0.723 **	0.062	0.166	0.117	−0.223	0.248	0.419	0.472 *
PL					0.106	0.321	0.451 *	−0.455 *	0.013	0.529 *	0.576 **
PBP						0.066	0.263	0.015	−0.232	−0.294	−0.303
SBP							0.548 *	0.065	−0.310	0.439	0.216
NFS								−0.625 **	−0.627 **	0.488 *	0.611 **
SS									0.421	−0.349	−0.646 **
TGW										0.033	0.004
HI											0.866 **

* Indicates significant at 5% level and ** Indicates significant at 1% level; DFF—days to 50% flowering; PH—plant height; NET—number of effective tillers per hill; FLL—flag leaf length; PL—panicle length; PBP—number of primary branches per panicle; SBP—number of secondary branches per panicle; NFS—number of filled spikelets per panicle, SS—spikelet sterility, TGW—1000-grain weight; HI—harvest index; GY—grain yield.

**Table 4 plants-11-02952-t004:** The phenotypic correlation coefficient among 12 characters in 20 genotypes of T. Aman rice.

	PH	NET	FLL	PL	PBP	SBP	NFS	SS	TGW	HI	GY
DFF	−0.035	0.016	−0.090	−0.257	−0.067	−0.109	−0.193	0.048	0.187	−0.212	−0.120
PH		0.201	0.718 **	0.644 **	−0.062	0.070	0.150	−0.292	0.384	0.497 *	0.580 **
NET			0.232	0.304	−0.177	0.254	0.617 **	−0.440	−0.233	0.501 *	0.635 **
FLL				0.647 **	0.059	0.163	0.116	−0.222	0.248	0.418	0.463 *
PL					0.128	0.302	0.398	−0.407	0.008	0.476 *	0.492 *
PBP						0.101	0.262	0.013	−0.221	−0.275	−0.291
SBP							0.538 *	0.062	−0.305	0.432	0.199
NFS								−0.620 **	−0.623 **	0.485 *	0.595 **
SS									0.421	−0.348	−0.634 **
TGW										0.033	0.004
HI											0.845 **

* Indicates significant at 5% level and ** Indicates significant at 1% level; DFF—days to 50% flowering; PH—plant height; NET—number of effective tillers per hill; FLL—flag leaf length; PL—panicle length; PBP—number of primary branches per panicle; SBP—number of secondary branches per panicle; NFS—number of filled spikelets per panicle; SS—spikelet sterility (%); TGW—1000-grain weight; HI—harvest index; GY—grain yield (t/ha).

**Table 5 plants-11-02952-t005:** Partitioning of genotypic correlation with GY into direct (bold) and indirect PCs in 20 genotypes of T. Aman rice.

	DFF	PH	NET	FLL	PL	PBP	SBP	NFS	SS	TGW	HI	GCGY
DFF	**0.018**	−0.001	−0.001	−0.015	0.007	0.013	0.019	−0.103	−0.011	0.049	−0.101	−0.125
PH	0.004	**−0.011**	0.006	0.110	−0.017	0.015	−0.008	0.085	0.061	0.106	0.237	0.587 **
NET	0.006	0.003	**0.015**	0.040	−0.004	0.035	−0.046	0.334	0.091	−0.056	0.238	0.656 **
FLL	0.001	−0.008	0.005	**0.150**	−0.019	−0.007	−0.030	0.062	0.046	0.070	0.202	0.472 *
PL	0.014	0.011	0.021	0.098	**−0.021**	−0.014	−0.041	0.196	0.086	0.019	0.208	0.576 **
PBP	−0.004	−0.002	−0.004	0.004	−0.009	**−0.209**	−0.036	0.137	−0.005	−0.056	−0.117	−0.303
SBP	0.003	0.004	0.007	0.029	−0.007	−0.028	**−0.202**	0.284	−0.007	−0.072	0.204	0.216
NFS	0.001	0.002	0.010	0.021	−0.009	−0.049	−0.103	**0.542**	0.125	−0.161	0.232	0.611 **
SS	−0.002	0.001	−0.007	−0.036	0.010	−0.005	−0.016	−0.331	**−0.200**	0.109	−0.169	−0.646 **
TGW	0.004	−0.005	−0.002	0.037	0.000	0.042	0.061	−0.333	−0.082	**0.268**	0.015	0.004
HI	0.001	−0.002	0.009	0.067	−0.011	0.055	−0.082	0.263	0.074	0.013	**0.479**	0.866 **

Residual effect = 0.13; * Indicates significant at 5% level and ** Indicates significant at 1% level; DFF—days to 50% flowering; PH—plant height; NET—number of effective tillers per hill; FLL—flag leaf length; PL—panicle length; PBP—number of primary branches per panicle; SBP—number of secondary branches per panicle; NFS—number of filled spikelets per panicle, SS—spikelet sterility, TGW—1000-grain weight; HI—harvest index; GCGY—genotypic correlation with grain yield (t/ha).

**Table 6 plants-11-02952-t006:** A list of the formula used to estimate different parameters studied in the experiment.

Attributes	Formula	Sources
Genotypic variance	σ^2^_g_ = (GMS − EMS)/r	[42]
Phenotypic variance	σ^2^_p_ = σ^2^_g_ + EMS	[42]
Environmental variance	σ^2^_e_ = EMS	[42]
Genotypic coefficient of variation	GCV (%) = $\frac{\sigma_{g}}{x}\times 100$	[48]
Phenotypic coefficient of variation	PCV% = $\frac{\sigma_{p}}{x}\times 100$	[48]
Heritability	h^2^_b_ = $\;\frac{\sigma_{g}^{2}}{\sigma_{p}^{2}}\times 100$	[49]
Genetic advance	GA = $\frac{\sigma_{g}^{2}}{\sigma_{p}^{2}}$ × k × σ_p_	[42]
Genetic advance a percentage of mean	GAPM = $\frac{\mathrm{Genetic}\;\mathrm{advance}}{\mathrm{Population}\;\mathrm{mean}}\times 100$	[42,50]
Genotypic correlation coefficient	r_g_ = $\frac{Cov_{g}1.2}{\sqrt{\sigma_{g}^{2}\;1\times {\;\sigma }_{g}^{2}\;2}}$	[42,52]
Phenotypic correlation coefficient	(r_p_) = $\frac{Cov_{p}1.2}{\sqrt{\sigma_{p}^{2}\;1\times {\;\sigma }_{p}^{2}\;2}}$	[42,52]

Notes: GMS = Genotypic mean square, EMS = Error mean square, r = Number of replications, X = Population mean, K = Selection differential, the value of which is 2.06 at 5% selection intensity, σ_p_ = Phenotypic standard deviation, *Cov_g_* 1.2 = Genotypic covariance of traits 1 and 2, *σ^2^_g_*1 = Genotypic variance of trait, and *σ^2^_g_2* = Genotypic variance of traits 2, *Cov_p_* 1.2 = Phenotypic covariance of traits 1 and 2, σ^2^*_p_* 1 = Phenotypic variance of trait, and σ^2^*_p_* 2 = Phenotypic variance of traits 2.

## Data Availability

All data generated or analyzed during this study are included in this published article.

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
