# Peer review of "Genetic Variability, Character Association, and Path Coefficient Analysis in Transplant Aman Rice Genotypes"

_plants, 2022, doi:10.3390/plants11212952_

Round 1

Author Response

Comment: The study on Genetic Variability, Character Association, and Path Coefficient Analysis in T. Aman Rice Genotypes is worthy of investigation and reporting. However, authors should revise thoroughly for the comments below;

Author response: Thank you for allowing us the opportunity to submit our revised manuscript for publication in the Plants Journal of MDPI. We appreciate the time and effort you have taken to improve our manuscript. We are also thankful to the honorable reviewer for the positive decision to publish in Plants. We revised our manuscript following your point-by-point comments and suggestions for substantial improvement. We hope that this revised version satisfies you to take the final decision.

Comment: 1- Authors have used only 20 breeding lines to evaluate the genetic deviation, trait association, and path coefficient for yield and yield related components. Also, author have not provided any details about these 20 germplasms taken in this study. Pedigree and other information’s related to genotype is missing even in the attached supplementary table.
Author response: Thank you for your comments. We added the pedigree and other information related to the genotypes used in this study in Supplementary Table S1. We modify the previous supplementary table S1 and the newly incorporated information is highlighted in green color.

Comment: 2- Are these 20 lines being sister lines evolved from same cross? Or they belong to different cross combinations? Authors could have mentioned all these details.
Author response: Thank you for your comments. No, they are not sister lines. They are the advanced lines that evolved from different crosses. All the related information is given in supplementary table S1.

Comment: 3- There are no checks used in this study. How without using any local or regional checks in set of 20 lines, authors gone for the breeding analysis and interpret the results and find the component traits contributing to the most complex trait such as yield?
Author response: Thank you for your comments. Previously we have conducted a performance trial. In our previous study, our all advanced lines (F7 generation) were grown and evaluated with a standard check (BRRI Dhan32). The lines that we found superior to the standard check are included in this study. Therefore, we did not include any checks for this study. We have the plan to conduct multilocation trials of the materials with standard local checks in the future.

Comment: 4- No figures are given in the manuscript. At least authors should include the PCA graphs and path analysis figures for easy understating to the readers.

Author response: Thank you very much for your valuable comments. According to your suggestions, we added the path diagram (Figure 1) and PCA graphs (Figure 2) in the revised manuscript.

Reviewer 2 Report

I am very happy to see the efforts made by authors

This will be of great work of interest for readers and researchers 

I have few minor suggestions for making this manuscript acceptable 

Points to be focused ……

- Grammar correction is required throughout the manuscript - 

-please improve the figure qualities 

- rewrite the conclusion 

- Discussion need to improvise 

Thank you 

Author Response

Comment: I am very happy to see the efforts made by authors. This will be of great work of interest for readers and researchers. I have few minor suggestions for making this manuscript acceptable 

Author response: Thank you for allowing us the opportunity to submit our revised manuscript for publication in the Plants Journal of MDPI. We appreciate the time and effort you have taken to improve our manuscript. We are also thankful to the honorable reviewer for the positive decision to publish in Plants. We revised our manuscript following your point-by-point comments and suggestions for substantial improvement. We hope that this revised version satisfies you to take the final decision.

Points to be focused ……

Comment: - Grammar correction is required throughout the manuscript - 

Author response: Thank you for your comments. We have thoroughly checked this manuscript with our university English expert team.

Comment: -please improve the figure qualities 

Author response: Thank you for your comments. According to the suggestion of you and reviewer 1, we included figure 1 (Path diagram) and figure 2 (PCA graphs) with high resolution.

Comment: - rewrite the conclusion 

Author response: Thank you for your comments. We have rewritten the conclusion accordingly

Comment: - Discussion need to improvise 

Author response: Thank you for your comments. We have tried our best to rewrite the discussion in the revised manuscript to make it more coherent and easier to follow.

Reviewer 3 Report

Dear authors,

the presented research gives a thorough analysis of basic selection tools in rice breeding. The manuscript will undoubtedly be useful for rice breeders and researchers, but there are some corrections to be made. All suggestions are in the uploaded file.

Best wishes and good luck!

Author Response

Comment: Dear authors,

the presented research gives a thorough analysis of basic selection tools in rice breeding. The manuscript will undoubtedly be useful for rice breeders and researchers, but there are some corrections to be made. All suggestions are in the uploaded file.

Best wishes and good luck!

Author response: Thank you for allowing us the opportunity to submit our revised manuscript for publication in the Plants Journal of MDPI. We appreciate the time and effort you have taken to improve our manuscript. We are also thankful to the honorable reviewer for the positive decision to publish in Plants. We revised our manuscript following your point-by-point comments and suggestions for substantial improvement. We hope that this revised version satisfies you to take the final decision.

Comment: Overall consideration: Please make sure all acronyms, abbreviations or initialisms are defined at the first mention in each of three sections: the abstract; the main text; the first figure or table. When defined for the first time, the acronym, abbreviation or initialism should be added in parentheses after the written-out form.

Author response: Thank you for your valuable comments. We have defined all the abbreviations for the first appearance of three sections: the abstract; the main text; the first figure or table in the revised manuscript.

Abstract:
Comment: Line 28: Please correct (h2b) to heritability in the broad sense (h2b), and GAPM to genetic advance as percent of mean (GAPM).

Author response: Thank you for your comments. The above-mentioned changes have been made to the revised manuscript.

Comment: Lines 30-35: The presented findings do not match those in the Conclusions.

Author response: Thank you for your comments. We have matched the abstract with the conclusion. However, we have reformed the writing in the conclusion to avoid repetition with abstract

Introduction:
Comment: Line 72: Please use the written-out form of GY with abbreviation in parentheses.
Author response: Thank you for your comments. We have used the written-out form of GY with the abbreviation in parentheses.

Comment: Line 76: Please use the written-out form of PL with abbreviation in parentheses.
Author response: Thank you for your comments. We have rewritten the elaboration the abbreviation of PL in parentheses.

Comment: Line 90: Please write all the abbreviations you are using for the first time in written-out form (DFF).

Author response: Thank you for your comments. We have defined all the abbreviations for the first appearance in the revised manuscript.

Results:
Comment: Line 107: Please write all the abbreviations you are using for the first time in written-out form (σ2p, σ2g, σ2e, GCV, PCV, h2b, GA, GAPM). Please use unified abbreviations throughout the text and tables (PV or σ2p, GV or σ2g).

Author response: Thank you for your comments. We have rewritten the written-out form of the above-mentioned abbreviations. We also unified the abbreviation for phenotypic variation (σ2p) and genotypic variation (σ2g) throughout the revised manuscript.

Comment: Line 117: Please correct PH to plant height (PH).

Author response: Thank you for your comments. We have corrected ‘PH’ as ‘plant height (PH)’.

Comment: Line 127: As GA for NET is very low (1.77), it should probably be mentioned, considering you mentioned GA for FLL (9.03) and PL (1.43).

Author response: Thank you for your valuable suggestions. According to your suggestion, we have included the GA value of NET in the proper section in the revised manuscript. Although GA is low but GAPM and heritability is high.

Comment: Line 128: Please use the written-out form of FLL with abbreviation in parentheses.
Author response: Thank you for your comments. We have changed according to your suggestion in the revised manuscript.

Comment: Line 134: Please use the written-out form of GV with abbreviation in parentheses.
Author response: Thank you for your comments. The above-mentioned modification has been written as ‘genotypic variance (σ2g)’.

Comment: Line 137: Please use unified abbreviations throughout the text and tables (NPB or PBP). As it is the first time this term is used, please use the written-out form with abbreviation in parentheses.
Author response: Thank you for your comments. The unified written-out form an abbreviation for “number of primary branches per panicle (PBP)” have been made throughout the manuscript.

Comment: Line 138: Please use unified abbreviations throughout the text and tables (NSB or SBP). As it is the first time this term is used, please use the written-out form with abbreviation in parentheses.
Author response: Thank you for your comments. The unified written-out form an abbreviation for “number of secondary branches per panicle (SBP)” have been made throughout the manuscript.

Comment: Line 139: Please use the written-out form of NFS with abbreviation in parentheses.
Author response: Thank you for your comments. We have defined the NFS with an abbreviation in parentheses in line 92 where it was first appeared in the text.

Comment: Line 142: Please use the written-out form of SS with abbreviation in parentheses.
Author response: Thank you for your comments. We have rewritten the written-out form of SS with the abbreviation in parentheses when it appears for the first time (line 92).

Comment: Line 144: Please correct TGW to 1000-grain weight (TGW).

Author response: Thank you very much for your valuable comments. We corrected TGW to 1000-grain weight (TGW).

Comment: Line 147: Please use the written-out form of HI with abbreviation in parentheses.

Author response: Thank you for your comments. We have rewritten the written-out form of HI with the abbreviation in parentheses when it appears for the first time.

Comment: Line 153: Please indicate that Table 1. contains mean sum of squares.

Author response: Thank you for your comments. The above-mentioned change has been made to the title of Table 1.

Comment: Lines 170 -172: Please include the reference to Tables 3 and 4 at the end of the sentence.
Author response: Thank you for your comments. We have included the reference at the end of the above-mentioned sentence.

Comment: Line 177: Please replace NPB with NSB.

Author response: Thank you for your comments. We have replaced the NPB with NSB

Comment: Line 178: Please replace NSF with NFS, IPC with NC.

Author response: Thank you for your comments. We have replaced NSF with NFS, IPC with NC.

Comment: Discussion: In the Discussion, you are commenting on the different analyses crucial for decision-making in selection process, in the same order as they were presented in the Results. However, there is no need for repeating the results. If the goal of the experiment was defining the parameters most useful for grain yield advance, it would be advisable to exclude less helpful parameters as you go. For example, first you comment on parameters based on their genetic diversity (GV-PV, GCV-PCV). Briefly, parameters with higher genetic variability and less environmental influence (PH, NET, PL, NFS, FLL, SS, HI) will facilitate the selection process, those with low variability can be useful, but the selection is limited. Among all of the parameters, those exhibiting low heritability and GA can be excluded (NET, FLL, PL, PBP, SBP, TGW, HI) so you are left with DF, PH, NFS and SS. At the end, among those parameters left over, you exclude those with low/insignificant GCGY and direct PC. The parameters left at the end are NFS and SS, and they should be the most efficient ones for indirect GY breeding. Other parameters with high correlation to GY but low GA can be used as well but are less efficient. Finally, you can comment on the relationship between NFS and SS with other parameters. For example, NFS is in a significant positive correlation with NET and NSB, which could be helpful if these parameters are more easy to observe during the vegetation or measure afterwards. I believe that if the Discussion was constructed in such a way, it would be easier to follow.

Author response: Thank you for your comments. We have deleted all results from the discussion part. For your kind consideration, we have no characters with low heritability. Again, except of PL, although some characters have low GA however, their GAPM is high (for selection, GAPM is more important than GA), so these traits could be selected for the advancement of rice yield. We followed your suggestion to revise the discussion.

Comment: Lines 368-370: What about NET?

Author response: Thank you for your valuable point. We have added the correlation of NET (r = 0.656) with GY.

Comment: Lines 377-379: Please elaborate on this conclusion so that it is more clear. If the goal is selection for higher grain yield, genotypes with lower DFF, NPB and SS will be favoured. If you are breeding for short-vegetation genotypes, or genotypes with multiple primary branches, then concurrent selection for grain yield can be a problem. However, the only SNC was between GY and SS, which is expected, and it should not be considered a problem, since selection for genotypes with higher spikelet sterility is pointless. 

Author response: Thank you very much for your comments. We would like to extend our gratitude to point out the weakness of the discussion. We have modified the above-mentioned section according to your suggestion.

Comment: Lines 392-395: You are repeating GCC results already mentioned in the paragraph above. It would maybe be better to focus on direct PCs here.

Author response: Thank you for your comments. We have changed the respective areas according to your suggestions.

Materials and Methods:

Comment: Line 422: Please remove the colloquial phrase ‘in the middle of nowhere’.
Author response: Thank you for your comments. We have removed the colloquial phrase ‘in the middle of somewhere ’.

Comment: Line 441: What do you mean by ‘important intercultural operation’? Is it inter-row cultivation?
Author response: Thank you for your comments. It was a transplanted rice cultivation system where rice was grown in a line maintaining row-to-row 20 cm and plant-to-plant 20 cm distance. The important intercultural operation includes gap filling, weeding at regular intervals, insecticide application, top dressing of fertilizers, etc.

Comment: Line 436: How many hills and rows were there after plants were transplanted?

Author response: Thank you for your comments. The experiment unit was a plot with a size of 5.0 m × 1.0 m, with a 20 cm × 20 cm spacing between rows and plants. Each genotype is made up of 5 rows, with 25 hills in each row. This gives each replication a total of 125 hills.

Conclusion:
Comment: Line 470: Please use unified abbreviations throughout the text and tables (PV or σ2p, GV or σ2g).

Author response: Thank you for your comments. We have unified the abbreviation of σ2p for genotypic variance throughout the revised manuscript.

Comment: Lines 476-479: You are singling out SS, NFS and HI only based on the direct PC, but what about TGW?

Author response: Thank you for your comments. Although the direct effect of TGW is high (0.268), we excluded this trait from our selection criteria due to the insignificant correlation (0.004) of GY with TGW (Table 5).

Comment: TGWs direct PC is higher than SSs. Nonetheless, PC analysis is not the only analysis relevant for decision-making in selection. For example, HI was chosen based on the direct PC, but its GA is low, indicating less success if use as a selection criterion. It is advisable to base your final conclusions on all of your analyses (see comments for the Discussion).

Author response: Thank you for your comments. Although the direct positive correlation of TGW is high (0.268), we excluded this trait from our selection criteria due to the insignificant correlation (0.004) of GY with TGW (Table 5). Additionally, from our PCA analysis, we found a negative contribution of TGW on the PC1and PC3 (Supplementary Table 2). Again, though GA for HI is low however GAPM is high so we select HI.
